# The Use of Social Media in Children and Adolescents: Scoping Review on the Potential Risks

**DOI:** 10.3390/ijerph19169960

**Published:** 2022-08-12

**Authors:** Elena Bozzola, Giulia Spina, Rino Agostiniani, Sarah Barni, Rocco Russo, Elena Scarpato, Antonio Di Mauro, Antonella Vita Di Stefano, Cinthia Caruso, Giovanni Corsello, Annamaria Staiano

**Affiliations:** 1Pediatric Unit, IRCCS Bambino Gesù Children Hospital, 00100 Rome, Italy; 2The Italian Pediatric Society, 00100 Rome, Italy; 3Department of Pediatrics, San Jacopo Hospital, 51100 Pistoia, Italy; 4Department of Translational Medical Sciences-Section of Pediatric, University Federico II, 80100 Naples, Italy; 5Department of Health Promotion, Mother and Child Care, Internal Medicine and Medical Specialties “G. D’Alessandro”, University of Palermo, 90100 Palermo, Italy

**Keywords:** social media, adolescents, children, social network, health, COVID-19

## Abstract

In recent years, social media has become part of our lives, even among children. From the beginning of COVID-19 pandemic period, media device and Internet access rapidly increased. Adolescents connected Internet alone, consulting social media, mostly Instagram, TikTok, and YouTube. During “lockdown”, the Internet usage allowed communication with peers and the continuity activities such as school teaching. However, we have to keep in mind that media usage may be related to some adverse consequences especially in the most vulnerable people, such as the young. Aim of the review is to focus on risks correlated to social media use by children and adolescents, identifying spies of rising problems and engaging in preventive recommendations. The scoping review was performed according to PRISMA guidelines, searching on PubMed the terms “social media” or “social network”, “health”, and “pediatrics”. Excluding articles not pertinent, we found 68 reports. Out of them, 19 were dealing with depression, 15 with diet, and 15 with psychological problems, which appeared to be the most reported risk of social media use. Other identified associated problems were sleep, addiction, anxiety, sex related issues, behavioral problems, body image, physical activity, online grooming, sight, headache, and dental caries. Public and medical awareness must rise over this topic and new prevention measures must be found, starting with health practitioners, caregivers, and websites/application developers. Pediatricians should be aware of the risks associated to a problematic social media use for the young’s health and identify sentinel signs in children as well as prevent negative outcomes in accordance with the family.

## 1. Introduction

Media device use is increasing year by year in Italy as well as in many other countries. An ISTAT report referred that in 2019, 85.8% of Italian adolescents aged 11–17 years regularly used smartphones, and over 72% accessed Internet via smartphones [1]. Almost 95% of Italian families with a child had a broadband internet connection [2]. Internet connection was mostly used to communicate with friends and to use social networks [1]. In 2020, COVID-19 pandemic represented one of the greatest disruptions for everybody’s everyday life, in Italy as well as all around the world. From the beginning of the pandemic period, media device and Internet access rapidly increased. In line, a 2021 CENSIS report revealed an even progressive increment of smartphone use by adolescents, which reached 95% [3]. In particular, the majority of adolescents (59%) admitted they use smartphone even more frequently than in the past with a daily use of more than 3 h in 46% of cases. Adolescents connected Internet alone (59%), consulting social media, mostly Instagram (72%), TikTok (62%), and YouTube (58%) [4]. In this context, social interaction over the Internet or simply social network consulting may play an important part in the lives of many young people, influencing them and their relationship with self-esteem and well-being [5]. Not being guided and monitored in Internet fruition, the young may be exposed to several risks, including cyberbullying which affects 7% of children aged 11–13 years and 5.2% of 14–17 years old adolescents or stalking which affects more than 600 minors in Italy. On social media, the young are more vulnerable and may display risk behavior, including pertaining substance abuse, sexual behaviors, or violence [6].

On the other hand, media and social networks are, actually, present in almost any house and are considered a great resource for anybody, including children and adolescents. Especially during “lockdown”, the Internet usage allowed communication with peers and the continuity activities such as school teaching. Social media services enable various form of communication verbally or visually by internet-based networking, bringing people together, facilitating instant connection and interaction, such as a like or a comment on something [7]. There was also a “school” use of smartphones and social media during lockdown which represented a tool of information and education [8].

In line, websites and applications that enable users to create and share content or to participate in social networking may be currently use as a definition of a social media. Facebook launched in 2004 and Twitter in 2006 were the first social media introduced, rapidly followed by many others [9]. Actually, Facebook with 2.9 billion monthly active users, YouTube with 2 billion, Instagram with 1.5 billion, and TikTok with 1 billion are the most accessed social media in the world [10]. As social media are spreading in every day’s life, regulatory models are required to address a broad range of challenges social media pose to the community, including privacy and protection of sensitive data.

Media usage is related to some adverse consequences especially in the most vulnerable people. The health emergency had a strong impact on the mental and psychological health of adolescents causing changing in their routine and daily activities. Forced isolation increased anxiety and stress especially in the most fragile individuals, such as children and adolescents, leading to a change in habitual lifestyles. The greatest risk was that of taking refuge in excessive use of smartphones, electronic devices, and social networks, running into a “digital overdose” [11].

A recent survey conducted by the Italian Society of Pediatrics in collaboration with State Police and Skuola.net investigated the relationship with media devices in times of pandemic, investigating the habits of adolescents on the use of media and social networks, underlined that 15% of them declared they “cannot stay without” their own media device [1].

The aim of the review is to focus on risks correlated to social media use by the young, identifying spies of rising problems, and engaging in preventive recommendations.

## 2. Materials and Methods

This scoping review has been conducted by The Italian Pediatric Society Scientific Communication Group in order to provide an overview of a complex research area. The aim is reviewing international literature disguising about social media and their effect on the pediatric age, including minors less than 18 years, to underline possible risks found so far, identifying the signs of a dangerous use, and to eventually give new recommendation based on these findings.

We define a risk as the possibility of something unfavorable happens, as an effect or an implication of social media usage and which may potentially affect human health. This scoping review has been performed according to the PRISMA Extension guidelines for Scoping Reviews [12].

An electronic search was undertaken on PubMed database on 23 January 2022. To avoid missing results that may be of note for our revision study, constructing our search in PubMed, we used all of the important concepts from our basic clinical question, avoiding unnecessary filters.

So, the search terms “social media”, “health”, and “pediatrics” in text or title/abstract were used, with the time span set as “all years”. The search on the selected database has produced *n* 651 among articles and reviews. Another research was made using “social network”, “health” and “pediatrics” as search terms in text or title/abstract, with the time span always set as “all years”. It resulted in 354 articles/reviews.

The two research were downloaded from PubMed and then uploaded to the web application “Rayyan” [13], a website used to screen and analyze articles, specific for writing reviews. Additional articles for potential inclusion were identified in a second stage by hand searching the reference lists in relevant articles.

Studies were considered eligible for this scoping review if they met the following inclusion criteria:-Full-length articles or reviews.-Pertaining to children and adolescents up to 18 years old.-Negative impact on a pediatric population using social media.-Social media meant as forms of electronic communication.

The exclusion criteria were:
-Reports not in English.-Duplications.-Not pertinent field of investigation (e.g., use of the social media to promote healthcare, benefits of social media, social media used to debate on health-related issues, and social network meant as real social interactions).-The population analyzed was adult (>18 years).-The population had previous pathologies.

To reduce errors and bias, two researchers independently, two researchers conducted the screening process to identify articles that met all inclusion criteria. Using the web application “Rayyan” [13], duplicates were removed, then titles and abstracts were analyzed to exclude distinctly irrelevant articles. Finally, the eligibility of the articles was confirmed by evaluating the full text. Disagreements regarding inclusion/exclusion were settled by discussion between the researchers.

Relevant articles were selected on the web application “Rayyan” and grouped together based on the issue they were dealing with. Afterwards, data were compiled in a Microsoft Excel spreadsheet to calculate frequencies and percentages of the problems related to social media use, found in the research.

## 3. Results

All the 1005 documents have been reviewed for relevance and eligibility.

As shown in the Figure 1, through the help of the web application “Rayyan” [13] we removed before screening 9 duplicates, 25 foreign language works, and 49 publications dated before 2004. We excluded paper published before 2004, the year of Facebook foundation, because before that year “social networks” was a term used to mean “social interactions in real life”, as they were not pertinent to our research.

According to PRISMA guidelines [12], of the 922 works identified, all abstracts were analyzed, and 832 records were excluded. Around 66% of the excluded records were dealing with other topics (e.g., vaccines, promoting health by social media, social networks meant as real social interactions, and social lockdown during SARS-CoV-2 period), a percentage of 28% of the records corresponded to a wrong population: mostly parents, pregnant women, young adults, or children with pathologies (e.g., ADHD). About 6% of the excluded studies used social media tools to recruit people in their studies or to deliver questionnaires.

In conclusion, 90 were the records to be analyzed reading their full-length articles. The whole article of four of them has not been found (“reports not retrieved”), arriving at 86 reports assessed for eligibility. Figure 1 presents the flow chart of the selection process, adapted from PRISMA guideline [14].

Of the 86 reports attained, we read the whole length articles and then excluded 20 studies.

Of these twenty, 6 were excluded because not leading to any conclusion; 13 were dealing with wrong topics, such as: doctors’ social media knowledge; social lock down during the pandemic; social media marketing; underage and privacy; survey on how social media is perceived by adolescents; time consumed on social media; predictor factors of problematic social media use. Finally, one was not included because it focused on parents and families.

Searching through the cited studies in the included reports, two reviews which were not initially included in the research were added.

With 68 included reports analyzed, there were 15 reviews; of these two were systematic reviews, one validation study, and one editorial. Cross-sectional studies and longitudinal studies have been considered, eight and nine, respectively.

Many articles reported more than one issue correlated to social media use. The most frequent problems involved mental health, followed by diet and weight problems. Table 1 shows the problematic topics found to be related to social media use in children and adolescents and their prevalence, expressed as percentage, over the 68 reports analyzed.

The most frequent problems found are related to mental health: depression, anxiety, and addiction.

Other problems are related to sleep, diet and nutrition, cyberbullying, psychological aspects, behavioral problems, sex, body image perception, physical activity, online grooming, sight, headache, and dental caries.

## 4. Discussion

### 4.1. Social Media and Depression

We identified 19 publications reporting a relationship between social media use and depression [15,16,17,18,19,20,21,22,23,24,25,26,27,28,29,30,31,32,33]. Table 2 summarized the main finding regarding each article. Out of them, four investigated the impact of COVID 19 pandemic on both social media use and depression (Table 2).

#### 4.1.1. Before COVID-19 Pandemic

Investigating the impact of social media on adolescents’ wellbeing is a priority due to a progressive increase in mental health problems or addiction and access to Emergency Department [15]. As Chiu and Rutter stated, there is a positive relationship between internalizing symptoms, such as depression and anxiety, and social media use [15,16]. Depression is connected to a rapidly increased of digital communication and virtual spaces, which substitute face-to-face contact by excessive smartphone use and online chatting. The more time adolescents spend on social device the higher levels of depression are found out. In this sense, social media are representing a risk factor for depression in the young. Depression, anxiety, and behavioral disorders are among the leading causes of illness and disability among adolescents [15,16,17,18,19,20,21,22]. Key findings which correlate to depression regarding social media exposure are repeated activities such as checking messages, investment, and addition [23]. The findings were similar all over the world.

For example, in Sweden, spending more than 2 h on social media was associated with higher odds of feeling [20]. In Egypt, as well, students who have problematic Internet use, have higher psychiatric comorbidities, such as depression, anxiety, and suicidal tendency [24].

Social media addiction and more precisely Facebook addiction was linked not only to depression but even to dysthymia, so that the expression “Facebook depression” was coined to identify a relationship between depression and social networking activity [15,25,26]. Individuals suffering from Facebook depression may be at an increased risk of social isolation and may be more vulnerable to drugs or behavioral problems [26].

Internet penetrance and connectivity are also connected to cyberbullying which can lead to depression and suicidality [27,28,29].

On the other side, physical activity may decrease depression and anxiety, potentially protecting the young against the harmful effect of social media abuse [16].

At last, even if a positive correlation between internalizing symptoms and media use device is noted, Hoge states that there is also evidence that social media communication may improve mood and promote health strategies in some occasions [18].

Finally, even if evidence revealed that social media use is linked to poor mental health, the relationship between social media and depression in adolescents is still to be completely understood. It is still unclear whether social media use leads to more depression or if these depressive symptoms cause individuals to seek out more social media, which could feed into a vicious cycle [16]. Keles’s conclusion as well suggest defining the relationship between internalizing symptoms and social media use as an association and not a causative effect [23].

#### 4.1.2. After COVID 19 Pandemic

During COVID-19 pandemic, the state of emergency and social isolation determined an increase in time on screen not only as a source of online education, but to continuously access social media. According to recent data, a percentage of 48% of adolescents spent a mean of 5 h per day on social media and 12% spent more than 10 h. Moreover, with that increase in virtual time depression arose [30].

The degree of social media usage in children is a significant predictor of depression, which increases with each additional hour of social media use [31].

During the pandemic, depressive symptoms may have been reactive to the context of being afraid of the virus and necessitating social isolation [32].

However, in this peculiar period, schoolchildren who increased time spent on either smartphones, social media, or gaming had significantly elevated psychological distress, such as depressive symptoms, than those with decreased time spent on these internet-related activities [33].

### 4.2. Social Media and Diet

Out of the reports, 15 dealt with the association of social media use and diet [21,34,35,36,37,38,39,40,41,42,43,44,45,46,47]. The problems were related to junk food marketing (9 reports) [34,35,36,37,38,39,40,41] obesity (4 reports) [21,41,42,43], unhealthy eating behaviors (3 reports) [44,45,46], and alcohol marketing (2 reports) [21,47]. In Table 3 the retrieved articles dealing with social media and diet, and their major findings are presented (Table 3).

#### 4.2.1. Before COVID-19 Pandemic

##### Junk Food Marketing

Reports found that children are exposed to the marketing of unhealthy foods on social media and to their persuasive techniques. Digital marketing represents a major threat for children and adolescents in Mexico, because of its persuasive techniques. Cola and soft drinks, sweetened juices and in general the so-called junk food have high followers on Facebook and Twitter. [34]. This may cause an increase in children’s immediate consumption of the promoted product, unhealthy behaviors and may led to obesity, as confirmed by several studies [34,35,36]. Reports agree on the youth major vulnerability to unhealthy food advertisement, including digital marketing, sponsored content, influencers, and persuasive design [34,35,36]. This contributes to the obesity epidemic [36]. 

Major social media platforms do not have comprehensive policies in place to restrict the marketing of unhealthy foods on their platforms [36,37]. Therefore, exposure to the marketing of unhealthy products, on social media may be considered a risk factor for related unhealthy behaviors.

Analysis of the advertising policies of the 16 largest social media platforms proved them ineffective in protecting children and adolescents from exposure to the digital marketing of unhealthy food [37].

Among social media, YouTube is particularly worrying considering the affinity of the young toward the platform. Unhealthy food advertisements predominate in YouTube content aimed towards children. In fact, analysis of advertisements encountered in YouTube videos targeted at children revealed that food and beverage ads appeared most frequently, with more than half of these promoting unhealthy foods [38].

As confirmed by an Irish study, adolescents are very attracted to junk food advertisements and are likely to share comments on their network: generalized linear mixed models showed that advertisements for unhealthy food evoked significantly more positive responses, compared to non-food and healthy food. Of all the advertising, they see in social media, they view unhealthy food advertising posts for longer [39]. This confirms the vulnerability of children towards ad and digital marketing.

Moreover, it has been demonstrated that adolescent heavy social media users (>3 h/day) are more willing to engage with food ads compared to light social media users, and are more willing to “like” Instagram food ads featuring many “likes” versus few “likes”, demonstrating the power of social norms in shaping behaviors. Adolescents interact with brands in ways that mimic interactions with friends on social media, which is concerning when brands promote unhealthy product. [40]. There is a need of more strict policies to limit digital marketing, which is becoming more and more intense, especially towards children and adolescents.

#### 4.2.2. After COVID-19 Pandemic

During the COVID-19 pandemic, this phenomenon even increased. In fact, the combination of staying at home, online education and social media usage have all caused screen time to surge and the food industry has been quick to identify this change in their target audience and has intensified online advertising and focused on children. The COVID-19 experience led to an increase in risk and severity of inappropriate behavioral eating habits, affecting the health and weight [41].

#### 4.2.3. Before COVID-19 Pandemic

##### Obesity

Social media is the first independent risk factor for obesity in primary school children and the second for high school students. In both primary school and high school models, children’s social media use has the highest impact on child’s BMI [42]. In addition, heavy media use during preschool years is associated with small but significant increases in BMI, especially if used ≥ 2 h of media per day [21].

#### 4.2.4. After COVID-19 Pandemic

Obesity and social media correlated through junk food advertisements [41,43]. During COVID 19 pandemic poor quality food, energy-dense, and nutrient-poor products consumption increased, leading to the risk of overweight and obesity. The phenomenon has been called “Covibesity” [41].

### 4.3. Unhealthy Eating Behavior

Some social media contents promote pro-anorexia messages [44,45,46]. These messages are no longer limited to websites that can be easily monitored, but instead have been transferred to constantly changing media such as Snapchat, Twitter, Facebook, Pinterest, and Tumblr. Consequently, pro-eating disorder content has become more easily accessible by the users. Pro-anorexia website use is correlated with a higher drive for thinness, lower evaluations of their appearance, and higher levels of perfectionism, and all correlates with eating disturbances [44,46].

In detail, there is a real bombardment of unhealthy messages on media promoting low-nutrition aliments and sugar-sweetened drinks [45].

It is likely that the suboptimal quality of online information on social media platform contributes to the development of unhealthy eating attitudes and behaviors in young adolescent internet users seeking nutritional information. They look for nutritional information on internet sources such as commercial websites or social media in order to lose weight. In this occasion, they may be exposed to higher risk of eating disorders due to the high quantity of misinformation. Moreover, they may find dangerous methods to rapidly lose weight with possible harm for their health [46].

Literature agrees on the risk of time spent on social media as well as on the poor quality and reliability of weight loss information on media [44,45,46].

### 4.4. Alcohol Marketing

Adolescents identify drinking brands to peculiar images of ideal adults. Brands know well this underlying psychological mechanism and promote that identity adolescents seek, with specific advertisement on social media [47].

Studies have shown that exposure to alcohol in TV or movies is associated with initiation of this behavior. The major alcohol brands have a strong advertising presence on social media, including Facebook, Twitter, and YouTube. Several studies underlined risky health behaviors, such as illegal alcohol use or overuse. Evidence suggests that peer viewers of this content are likely to consider these behaviors as normative and desirable. Therefore, targeted advertising via social media has a significant effect on adolescent behavior [21].

### 4.5. Social Media and Cyberbullying

We identified 15 publications reporting a relationship between social media use and cyberbullying [21,22,25,26,27,28,29,45,48,49,50,51,52,53,54]. Table 4 summarized the main finding regarding each article (Table 4).

Cyberbullying may be defined as any behavior performed through electronic or digital media by individuals or groups that repeatedly communicate hostile or aggressive messages intended to inflict harm or discomfort on others. Compared to bullying, cyberbullying may be even more dangerous as victims can be reached anytime and in any place. Moreover, anonymity amplifies aggression as the perpetrator feels out of reach.

Moreover, the ability to hide behind fake names provides bullies the opportunity to communicate in content and language they would not use in front of people [26,48,49]. As confirmed by Shah et al., the anonymity of cyberbullying increases the risk for inappropriate behaviors among adolescents [50].

In literature, cyberbullying has been identified in phone calls, text messages, pictures/video clips, emails, and messaging apps. This is a great public health concern: in Italy, 2015 ISTAT data showed that 19.8% of 11–17 years old internet users report being cyberbullied [49].

This phenomenon is increasing. In fact, the number of adolescents being cyberbullied at least once in their life increased from 20.8% in 2010 to 33.8% in 2016 [50].

Victims of bullies exhibit increased depressive symptoms, anxiety, internalizing behaviors, externalizing behaviors, and greater academic distractions [21,22,25,27,28,29,51].

Cyberbullying has been associated with higher risks of depression, paranoia, anxiety, and suicide than the traditional form of bullying [21,22]. According to a metanalysis of 34 studies, traditional bullying increased suicide ideation by a factor of 2.16, whereas cyberbullying increased it by a factor of 3.12 [39].

In adolescence, social media intense or problematic use and frequent online contact with strangers are all independently associated with cyberbullying [45,52,53]. In this contest, social media represent a risk factor for cyberbullying and for inappropriate behavior related to it. In fact, problematic social media use is an important driver of cyberbullying victimization and perpetration, especially among girls [50,53]. The highest percentage is observed in adolescents, aged 13 to 15 years as suggested by literature reviews and, in particular, by Marengo and Uludasdemir [53,54]. However, Marengo also suggests that in presence of social support, the phenomenon is attenuated [53].

Moreover, having daily access to the Internet and the sharing of gender on social media increased the likelihood of cyber victimization among adolescents aged 12–17 years. Those who use Tumblr and Snapchat were found to become victims even more frequently [54].

### 4.6. Psychological Problems and Social Media

We identified 14 publications reporting a relationship between social media use and psychological problems [17,23,33,45,49,52,55,56,57,58,59,60,61,62]. Table 5 summarized the main finding regarding each article (Table 5).

#### 4.6.1. Before COVID-19 Pandemic

A high use of screen device has been correlated to a low psychological well-being among children and adolescents, especially among females [17].

For examples, in Canadians adolescents, the prevalence of loneliness was higher for daily computer-mediated communication users than non-daily users [55]. As well as for cyberbullying, adolescents may benefit from social support, family communication, and interaction to ameliorate feelings of loneliness [53,55]. Boer et al. confirmed that intense user reported more frequent psychological complaints than non-intense user as well as less family and friend support [56]. In line with this finding, in Lithuania a problematic social media use has been associated with two times higher odds for lower life satisfaction [57].

Moreover, an intense social media use correlated to either low school well-being and reduced social well-being (decreased family and friends support and relations) [56].

A relationship between poor life satisfaction, problematic social media use, and lack of social support was found not only in adolescents, but also in children [52,57,58,59,60].

Social media use is also correlated with conduct and emotional problems, attention deficit, peer problems, school impairments, and psychological distress [23,45,61,62].

Social networks and media device use correlate to low academic outcomes, reduced concentration, and procrastination. In fact, problematic smartphone use correlates to a surface approach to learning rather than to a deep approach, leading to reduced creativity, organization skills, own thinking, and comprehension of information [49].

#### 4.6.2. After COVID-19 Pandemic

During this COVID-19 pandemic, primary school children reported significantly higher psychological distress than the period prior to the COVID-19 outbreak. Studies showed that schoolchildren who increased time spent on either smartphones, social media, or gaming had significantly elevated psychological distress than those with decreased time spent on these internet-related activities [33].

### 4.7. Social Media and Sleep

Extended use of digital media screen time correlates with sleep impairment [18,21,22,26,31,43,47,49,57,61,63,64,65]. Table 6 summarizes the evidence in literature (Table 6). Exposure to screen-based devices, online social networking sites, and video-sharing platforms is significantly associated with sleep-onset difficulties in adolescents [18,49]. Findings from a meta-analysis of 20 cross-sectional studies show 53% higher odds of poor sleep quality among adolescents with consistent bedtime media use [63]. Moreover, the use of computers and smartphones among adolescents is associated with daytime sleepiness and fatigue, shorter sleep duration, later bedtime, and unfavorable changes in sleep habits over time [22]. Smartphones may be easily carried around and even taken to bed. Several sleep disorders correlate to both overall and night phone use among adolescents. It has been demonstrated that social media addiction in school students decreases students’ sleep efficiency [61]. Use of cellphones, particularly for nighttime texting, and consulting social media were associated with insufficient sleep [63]. A 5 or more hours daily of media devices use has been related to a higher risk of sleep problems when compared to a 1 h use daily [49]. This finding is confirmed by Buda who correlates problematic social media with about two times higher odds for a bad sleep quality [57]. Varghese as well associated social media use with sleep difficulties. Furthermore, YouTube user had two-times higher odds for sleep-onset difficulties [63].

In addition, it seems that girls suffer more than boys from these sleep problems [57].

Sleeping problems, especially sleep duration, have been then associated with time spent on screen, problematic behaviors, and higher internalizing and externalizing symptoms [64].

Even among children, there is a problem with extended use of social media sites, which result in sleep deprivation due to delayed bedtimes and reduced total sleep duration and quality of rest [31,65]. The report by Hadjipanayis as well confirms that sleeping disturbances may be associated with the disruption of circadian rhythms due to the blue light emission from the electronic screen-based media devices [26]. Negative outcomes including poor school performance, childhood overweight and obesity, and emotional issues have all been associated with sleep deprivation [21,26,43,47]. Inadequate sleep quality or quantity associated to social media use represents a risk factor for metabolic conditions such as for diabetes, cardiovascular disease and for mental problem, such as depression or substance abuse [49].

### 4.8. Social Media and Addiction

Ten reports found correlations between social media use and risk of different types of addictions: with internet [17,24,49,51,52,66], with substance abuse [15,67], with alcohol addiction and gaming [67], with gambling [68], and with tobacco use [69]. In Table 7, the major findings of the related reports are presented (Table 7).

Investigating the impact of social media on adolescents’ wellbeing is a priority due to a progressive increase in mental health problems and access to Emergency Department [15]. Chiu reported that mental health or addiction related emergency department access increased by almost 90% in ten years mainly among adolescents aged 14–21 years. The increment well correlates to an increase availability of social media [15].

High screen use associated with internet addiction is also confirmed by O’Keeffe who states that technology is influencing children’s lives from a very young age [51].

More than 7% of youth have problematic social media use, indicated by symptoms of addiction to social media [52]. Warning signs of internet addiction can be skipping activities, meals, and homework for social media; weight loss or gain; a reduction in school grades [41]. In detail: concern, loss of controlling tolerance, withdrawal, instability and impulsiveness, mood modification, lies, and loss of interest have been identified as risk factors for smartphone addiction. Females have almost three times more risk for smartphone addiction than males and it may be related to a stronger desire for social relationships [66]. Main problems correlated to addiction are low self-esteem, stress, anxiety, depression, insecurity, solitude, and poor scholastic outcomes. Smartphone addiction correlates to both fear of missing out (FOMO) and boredom. FOMO is the apprehension of losing experiences and the consequent wish to remain constantly connected with others, continuously checking social applications. Boredom is defined as an unpleasant emotional state, related to lack of psychological involvement and interest associated with dissatisfaction, to cope with boredom adolescents may seek additional stimulation and compulsively use smartphones [49].

As well as O’Keeffe, Hawi found out that children are starting to use digital devices at a very young age, and so should be screened for the risk of digital addiction. New scales of early identifications have been developed such as the Digital Addiction Scale for Children, validated to assess the behavior of children 9 to 12 years old in association with digital devices usage. Out of the sample size, 12.4% were identified as at risk of addiction and most of them (62.4%) were male. Nevertheless, results demonstrated that weekday device use among females causes more conflicts [66].

Different grading scales can test addictions. A study assessed 700 adolescents aged from 14 to 18 years and found out that 65.6% were having internet addiction, 61.3% were gaming addicts, and 92.8% Facebook addicts. Internet addict students had statistically significant higher age, higher socioeconomic scale score, male gender, and lower last year grades in comparison to non-addicts. Depression, dysthymia, suicide, social anxiety, and phobias were common comorbidities in addicted adolescents [24].

In undergraduate students, disordered online social networking use is associated with higher levels of alcohol craving and in pupils aged from 11 to 13, it is associated with a higher likelihood of being substance users [67]. In addition, excessive video gaming is associated with increased substance use [15,67].

One report showed greater risk for children and adolescents to develop gambling problems. In fact, the prevalence of adolescent gambling has increased in recent years. Across Europe, self-reported rates of adolescent gambling in 2019 ranged from 36% in Italy to 78% in Iceland. Adolescent problem gambling prevalence ranges from 1.6 to 5.6%. Not only adolescents but also children are widely exposed to gambling advertisements on television and via social media. In recent years, there has been an expansion in sports betting online, and this has been heavily promoted by advertising and marketing attractive to adolescents. Gambling is also promoted to children via social media: children are sharing and re-tweeting messages from gambling companies, they are active in conversations around gambling, and regularly consume and share visual gambling adverts. Lastly, there is also a strong relationship between gaming and gambling: in video games, children pretend to gamble and some video games would ask real money to play [68].

Finally, there might be a relationship between youth using tobacco and tobacco social media posts. It is not clear if the relationship can be cause-effect or only a correlation. Adolescents who participate in conversations about tobacco in social media by posting positive messages about tobacco are more likely to be past-month tobacco users. Posting even only one positive tobacco-related tweet was associated with greater odds of using cigarettes, e-cigarettes, or any tobacco product, compared to those who did not post positive messages about tobacco [69].

Finally, social media has been associated to social media use and may represent a risk factor for the young as it interferes with dailies activities leading to unhealthy habits. The easy access to social media by smartphone undoubtedly facilitates addiction.

### 4.9. Social Media and Anxiety

We identified 10 publications reporting a relationship between social media use and anxiety. Out of them, three investigated the impact of COVID 19 pandemic on social media use and anxiety [15,16,17,18,22,23,31,32,33,70]. Table 8 summarized the main findings (Table 8).

#### 4.9.1. Before COVID-19 Pandemic

Evidence agrees that the degree of social media usage in children is a significant predictor of anxiety and perceived stress levels and that it increases with each additional hour of social media use [17,23,31]. Anxiety may represent a risk factor for children and adolescents’ health as it influences the way they see their body, the way they feel, and it may impact on social acceptance and relations with peers.

The excessive use of at least one type of screen, including television, computer, social media, and video gaming, has been connected with anxiety symptoms in the pediatric age [22,23,31]. Furthermore, in Rutter’s study a significant association between depression and anxiety with social media use has been detached [16]. Nevertheless, it is still unclear if social media use provoke anxiety or if anxiety is the cause of excessive use of social media [16]. Emergency department visits for mental health, including anxiety problems, has arisen since 2009, likely linked to the increased use and the harmful effect of social media [15]. On the contrary, physical activity may protect the young against the harmful effect of social media, preventing depression and anxiety [16].

In a scientific report, Muzaffar confirmed that an association between anxiety and social media is of note. In detail, increased adolescent generalized anxiety symptoms were associated with increased Facebook use and repetitive Facebook habits. Anxious adolescents may not be able to control their discomfort to the point that they need to regularly go back to check their previous posting on Facebook [70].

The constant connection to social networks through digital devices, on its side, potentially contributes to feelings of anxiety. Adolescents and children suffering from social anxiety may prefer to interact with texting, instant messaging, and emailing than over face-to-face interactions. However, the behavior may increase risk in individuals vulnerable to social anxiety disorder because substituting digital media for interpersonal communication to avoid feared situations may be reinforced over time, making the person even more avoidant and worsening the symptoms and severity of social anxiety disorder [18].

However, in some studies, not just overexposure but also underexposure to social media was associated with adolescent anxiety, depression, and suicidal ideation [22].

#### 4.9.2. After COVID-19 Pandemic

Screen time and social media use have increased during the pandemic. Social media has been helpful during lockdown to keep social relationships and not to discontinuate school activities. However, an excessive Internet use may negatively affect children and adolescents’ well being. So, during social lockdown, an elevated psychological distress and anxious symptoms have been described in schoolchildren who increased time spent on screen [32,33]. Children who increased by 15 or 30 min daily the time spent on internet presented a high level of psychological distress.

### 4.10. Social Media and Sex Related Problems

Studies have found social media use related to sexual problematic behaviors such as early sexual activity, exposure to pornography, and sexting. [21,22,26,50,51,71,72,73,74]. Table 9 summarizes the results (Table 9).

The prevalence of sex related problems cannot be accurately recorded as for a wide range of definition and sampling methods and the comparison among reports is difficult.

Especially for girls, higher social media use, associated with lower family affluence and poorer body image, are key to early sexual activity [71].

Social media use was found to be significantly associated with risky sexual behavior among pre-college students in Ethiopia. Facebook, Instagram, YouTube, and other platforms have been identified as a factor that alters adolescent’s perception and influences them to engage in risky sexual behavior. Those who view sexually suggestive Facebook photos have a higher chance of having unprotected sexual intercourse and sex with strangers [72].

Moreover, youth can be exposed to unwanted sexual material online, including unwanted nude pictures or sexually explicit videos through means such as pop-up windows or spam e-mails [73].

Children exposed to inappropriate sexual content are prone to high-risk behaviors in subsequent sexual encounters. [22] Sexting activities may also affect emotional and social wellbeing of adolescents; it is correlated to depression and risky health behaviors, such as substance use, alcohol consumption, and suicide [26,50]. The odds of risky sexual behavior were 1.23 higher in social media user than in other students [72]. Furthermore, on the internet, pornography is readily accessible by media device, so that Wana found out that 7% of students use social media for pornography. In most cases, adolescents admit they intentionally viewed materials [74]. Pornographic media depict a fantasy world in which unrealistic encounters result in immediate sexual gratification, and intimate relationships are nonexistent. Repeated exposure of the adolescent brain to the world of online pornography can make it difficult for adolescents to develop mature healthy sexual relationships [22].

Internet pornography usage has been documented in adolescents before the age of 18. Online pornography is often the first source of sex education for many adolescents, and exposure to violent pornography increases the odds of sexually aggressive behavior [50]. Peer advice as well as substance abuse are significant predictor for risky sexual behavior [72].

Finally, among adolescents 10–19 years of age, the rate of sexting ranges from 5 to 22% [50,72,74].

Sexting is the use of media to send nude or sexualized contents such as texts, photos, or videos. An extensive sharing of these contents through technology has been connected with a negative impact on the emotional and social wellbeing of adolescents involved. An earlier sexual debut such as the use of drugs and promiscuity have been all associated to the excessive use of sexting. It can also cause spreading of sexual content material without consent, to a third party as a method of bullying or revenge [21,26,51,74].

### 4.11. Social Media and Behavioral Problems

Out of the reports, seven explored the influence of social media and behavioral problems [22,49,64,75,76,77,78]. Table 10 outlines the highlighted findings (Table 10). Behavioral outcomes usually cover five areas, including hyperactivity/inattention, emotional symptoms, conduct problems, peer relationship, and pro-social behavior.

For children aged 10–15 years old, limited time on social media has no effect on most emotional and behavioral outcomes (and can even positively impact social relationships), while there are strong negative associations between very long hours on social media and increased emotional distress and worse behavioral outcomes, which continue for several years [75].

In accordance to McNamee, the study by Okada conducted in Japan [76] among children aged 9–10 years old highlighted that mobile devices usage time of less than 1 h was a protective factor for behavior problems in boys. Instead, the usage time of 1 h or more was a risk factor in girls. Among girls, a dose–response positive association was found between duration of mobile devices usage and total difficulty score. A U-shaped association was found between duration of mobile devices usage and behavioral problems in boys: moderate use of mobile devices might be a tool for relaxation or alleviating distress through interactions with peers. However, in the subscale analysis, boys who use two or more hours of mobile devices showed higher risk of emotional problems and peer problems [76].

Moreover, the social media violent content exposure may be a risk factor for violent and aggressive behaviors. In this context, levels of aggression are directly proportional to exposure of types of violent media content. Electronic and social media showing contents with fights, stealing, dead bodies, and people’s belongings being destroyed influence young viewers, as per observational-learning theory, making them believe that reacting aggressively in response to perception of any offense is acceptable [77].

In line with Tahir’s report, Maurer underlined a significant association between exposure to media violence and aggressive behavior, aggressive thoughts, angry feelings, and physiologic arousal. Media exposure is also negatively related to personal adjustment and school performance and positively related to risk-taking behaviors [22].

Another study confirmed that longer the time spent on screens, higher the risk for behavioral problems among children 9–10 years old, and depending on the content type visualization, the risk for an aggressive and rule-breaking behavior. This association was mediated by sleep duration: longer sleep duration was associated with fewer problem behaviors [64].

Challenges and risk-taking attitudes are frequent in child and youth culture. However, online challenges take on new meanings when mediated by digital sociability; they appear as a powerful communicative resource to reaffirm belonging, recognition, and audience adherence. They are a media strategy adopted by youth in the construction of an internet-mediated identity in which risk and violence are crucial devices in building a self-image capable of maintaining an audience. Nevertheless, they can involve potential self-inflicted injuries to participants, with risks ranging from minor to even lethal [78].

Finally, an emerging problem is the social phenomenon called Shakaiteki Hikikomori (social withdrawal). Most of them are males and they usually experience a social reclusion range from 1 to 4 years. They refuse to communicate even with their own family and spend even more than 12 h a day in front of a screen [49].

### 4.12. Social Media and Body Image

On social media platforms such as Facebook, Snapchat, and Instagram, body image has become an important topic [17,25,45,46,50,73]. Table 11 summarized the evidence. (Table 11). People post their most flattering photos and view those of others, creating an online environment that could be damaging to body image acceptance. Spending time on social media puts adolescents under a higher risk of comparing themselves to models that are more attractive. As a result, these unfavorable social comparisons of physical appearance may exacerbate body image apprehension [17,45].

Moreover, beauty trends are constantly reinforced through social media networks and image-editing tools are often used to alter images to fit beauty standards. Teenagers who, perhaps, are not aware of these digital changing made in commercial photos may become insecure of their image. This may reduce self-esteem and body satisfaction, mainly among adolescent girls, developing body image concerns, engaging in weight-modification behavior, and potentially developing eating disorders. Nowadays, adolescents, and, in particular, girls, need to fit “social media” standard for photo posting; they use to modify photos with specific programs in order to respect society beauty standard. In fact, 28% of girls aged 8–18 years admit to editing their photos to make themselves look more attractive prior to posting online [50].

In addition to social media causing body image problems, adolescents with body image misperception may look on the internet for advice on how to lose weight quickly. However, the suboptimal quality of online information contributes to the development of unhealthy eating attitudes and behaviors in young adolescents. It may be that the content of these sites promotes eating disorders by providing unhealthy weight loss advice [46].

Furthermore, the desire of perfection and selfie mania with repeated selfie can cause depression and self-harm. This is a typical symptom of body dysmorphic disorder [73].

Finally, this association between the use of social media, self-esteem and body image can be a correlation and not a cause-effect relation: girls with lower self-esteem and sensitive to body image complains may use social media more frequently than girls with a higher level of self-esteem. For example, users can make a “selective self-presentation” where they show themselves only in a positive way on their social media profiles [25].

### 4.13. Social Media and Physical Activity

Evidence supports a correlation between social media and physical activity [45,49,57,73,79]. Excessive use of smartphones and other digital devices can also cause physical problems, such as a more sedentary lifestyle [45], which is positively associated with childhood obesity. In addition, non-physiological postures assumed while using smartphones may lead to cervical rigidity and muscle pain resulting in neck strain or “Tech Neck”. Moreover, “texting thumb” is a form of tendinitis that comes from overusing the thumb from excessive texting, video gaming, and web browsing using a smartphone [49,73].

An Australian study found that non-organized physical activity declines between 11 and 13 years, especially in children with a large increase in activities of texting, emailing, social media, and other internet use [79].

Another study showed that problematic social media use is related to lower levels of vigorous physical activity, especially in girls [57].

In Table 12 are listed the reports related to this topic and their major content (Table 12).

### 4.14. Online Grooming

Online grooming may be defined as a situation in which an adult builds a relationship with a minor finalized to a sexual abuse using social media. [47,80]. The risk of developing post-traumatic stress disorder in the victims is of note and may affect mental and well-being of children and adolescents [80].

Children are more vulnerable online as they often escape their parents’ control and may be more willing to share information or pictures about themselves than in real life.

Online grooming, differently to offline sexual abuse, is simpler to perpetrate, due to internet’s technology and accessibility. Furthermore, often the perpetrator misrepresents himself as another child or teenager, in order to establish a trusting relationship [21].

Teenage girls appear to be more at risk, even if the proportion of male victims is considerable too. In general, minors with problematic internet use are at greater risk of being groomed.

Sexual solicitation has been found to be more common in children spending longer time on internet on weekdays, being involved in sexting, having strangers in social networks friends list, playing online games, and chats. The risk is high even for adolescents whose curiosity and unconsciousness set them at risk of being deceived [80].

Table 13 presents the reports related to this topic and their major content (Table 13).

### 4.15. Social Media and Sight

Studies have investigated the risk of social media on sight, in terms of visual imbalance [22,49,73]. Evidence underlines that children can develop ocular disorders from excessive screen time, including myopia, eye fatigue, dryness, blurry vision, irritation, burning sensation, conjunctival injection, ocular redness, dry eye disease, decreased visual acuity, strain, fatigue acute acquired concomitant esotropia, and macular degeneration. During smartphone use, there is a reduction of the blink rate to 5–6/min that promotes tear evaporation and accommodation, leading to dry eye disease [49,73].

In addition, excessive screen time and less time spent outdoors may lead to early development of myopia, particularly with smartphone and tablet use [22].

Table 14 presents the reports related to this topic and their major content (Table 14).

### 4.16. Social Media and Headache

There are increased complaints of headaches related to staring at a screen for too long [62,73,81]. Reports dealing with social media and headache are listed in Table 15 (Table 15).

Headache is actually the most common neurologic disorder in the population, children and adolescents included [81]. It may negatively impact on children and adolescents’ well-being, leading to stress, tiredness, anxiety, and bad mood. Time of usage of media device has been directly connected to headache: in particular, adolescents using more than 3 h a screen have a significantly higher risk of headache compared with those using a device for less than 2 h (*p* < 0.001). Spending even 2–3 h with a computer significantly increases the chance of suffering a headache in comparison with those using a computer for less than 2 h (*p* < 0.01). Excessive use of electronic devices is considered a risk factor, especially for the development of migraine-type headache (*p* < 0.05) [81].

According to recent studies, headache and somatic symptoms have been found mostly in patients with problematic social media usage, compared with non-problematic peers. There is a consistent association between the problematic use of social media and adolescent psychosomatic health [62,73].

### 4.17. Social Media and Dental Caries

The association between use of internet and social media has been studied in literature [35,82]. Table 16 summarizes the main findings (Table 16).

The association between use of internet social media to obtain oral health information and dental caries has been highlighted in Almoddahi’s report [82]. In detail, problematic internet use has been associated with unhealthy lifestyles, poor oral health behaviors, and more oral symptoms such as toothache, bleeding gums, and poor self-perceived oral health. Caries and junk food have been both connected to excessive internet use and ads [82]. Therefore, social media may be a risk factor for caries, poor oral health, and dental outcomes.

In line with Almoddahi, Radesky underlines that advertisements on social media promote intake of foods that contribute to dental caries, such as fast food and sugar beverages [35]. Nevertheless, evidence suggests that smartphone applications may improve health and oral health when internet-based health interventions are in place. Delivering oral health information via social media may increase tooth brushing and dental outcome [82].

## 5. Limitations

From the literature, it is not possible to decide whether social media use causes internalizing symptoms and problematic behaviors examined in this manuscript or whether children and adolescents suffering from depression, anxiety, or other psychological distress are more likely to spend time on social media. We can just state that there is an association between social media use and health problems, but that is not necessarily cause-effect. Moreover, the articles included are different, ranging from reviewers to clinical studies to letters and to editors, so that it may be difficult to accurately compare them. Third, as specified in the materials and methods, we excluded reports not in English letter and not published in PubMed.

Nevertheless, through our manuscript we contribute to the existing literature to highlighting the impact of social media use on adolescents, providing advices to pediatricians in everyday practice.

## 6. Conclusions

Social media is increasingly being used by children and adolescents, especially during COVID-19 pandemic and the health emergency. Although social media use demonstrated to be of utility, an excessive or non-correct use may be a risk factor for mental health, including depression, anxiety, and addiction.

Social media use may also correlate to a non-adequate nutrition with consumption of junk food marketing leading to weight gain, obesity, dental caries, and unhealthy eating behaviors. Associations have been found also with increasing physical problems due to sedentary lifestyle, obesity, and non-physiological postures. On the other hand, social media can cause problems with body image visualization and acceptance, especially in young adolescent girls with lower self-esteem, who may look for contents for losing weight rapidly, and this can help the extension of anorexia disorders.

Children and adolescents who use social media for many hours a day, are also at higher risk for behavioral problems, cyberbullying, online grooming, sleep difficulties, eye problems, (such as myopia, eye fatigue, dryness, blurry vision, irritation, burning sensation, conjunctival injection, ocular redness, and dry eye disease), and headache. Moreover, uncontrolled social media use, can lead to sexting, exposure to pornography, exposed to unwanted sexual material online, and early sexual activity. Social media users meet more online risks than their peers do, with an increased risk for those who are more digitally competence.

Public and medical awareness must rise over this topic and new prevention measures must be found, starting with health practitioners, caregivers, and websites/application developers. Families should be educating on the dangers and concerns of having children and adolescence online. Prerequisite to inform families how to handle social media is to educate those responsible for training, including health practitioners. In detail, pediatricians should be reminded to screen for media exposure (amount and content) during periodic check-up visits. They need to keep in mind a potential correlation of problematic social media use with depression, obesity and unhealthy eating behavior, psychological problems, sleep disorder, addiction, anxiety, sex related problem, behavioral problem, body image, physical inactivity, online grooming, sight compromising, headache, and dental caries. Pediatricians can also counsel parents to guide children to appropriate content by consulting ratings, reviews, plot descriptions, and by a previous screening of the material. They should inform parents about the potential risk of digital commerce to facilitate junk food, poor nutrition and sweetened aliments, facilitating overweight and obesity. On the contrary, a healthy diet, adequate physical activity and sleep need to be recommended. Pediatricians may also play a role in preventing cyberbullying by educating both adolescent and families on appropriate online behaviors and on privacy respect. They should also promote a face-to-face communication and to limit online communication by social media. Pediatricians may encourage parents to develop rules and strategies about media device and social media use at home as well as in every day’s life.

## Figures and Tables

**Figure 1 ijerph-19-09960-f001:**
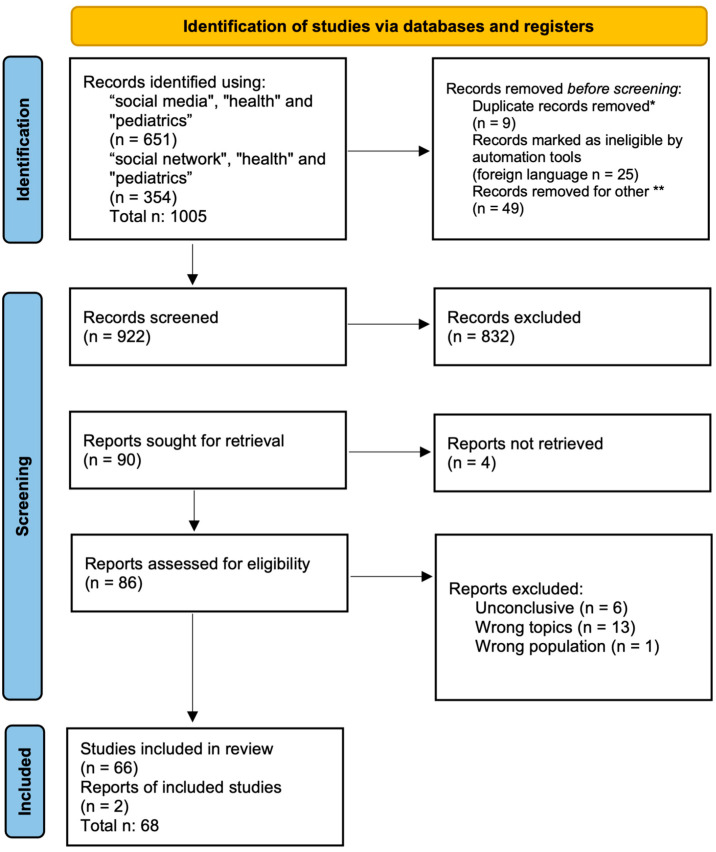
Flow chart of the selection process. * automation tools were used: 6 records were excluded by automation tools and 3 were excluded by authors. Twenty-five records were excluded because they were not written in English, these were identified using automation tools, but then checked by authors. ** 49 records were removed because they were published before 2004, and no social network existed before that year.

**Table 1 ijerph-19-09960-t001:** Social media health related problems in a pediatric population. This table shows the issues found in this scoping review. Depression was argued in 19 reports, being the main topic found (27.9% of the whole study). Diet associated problems were discussed in 15 reports, cyberbullying in 15, psychological problems in 14, sleep related problems in 13, addiction in 10, anxiety in 10, sex related problems in 9, behavioral problems in 7, body images distortion in 6, reduced physical activity and related problems has been reported in 5 reports, online grooming in 3 reports, sight problems in 3, also headache in 3, and dental caries in total of 2 articles.

Issue	*n*	%
Depression	19	27.9%
Diet	15	22.1%
Cyberbullying	15	22.1%
Psychological Problems	14	20.6%
Sleep	13	19.1%
Addiction	10	14.7%
Anxiety	10	14.7%
Sex Related	9	13.2%
Behavioral Problems	7	10.3%
Body Image	6	8.8%
Physical Activity	5	7.4%
Online Grooming	3	4.4%
Sight	3	4.4%
Headache	3	4.4%
Dental Caries	2	2.9%

**Table 2 ijerph-19-09960-t002:** Social media and depression.

Domains	Reference	Type of Publication	Highlighted
Depression Addiction Anxiety	Chiu M. et al. [15]	Clinical study	Social media increased use correlates to Emergency Department visits for mental illness, including depression, addiction, and anxiety.
Depression Anxiety	Rutter L.A. et al. [16]	Clinical study	Social media use correlates with depressive symptoms, anxiety, and loneliness. Physical activity negatively correlates with depression.
Depression Psychological problems Addiction Anxiety Body image	Mougharbel F. et al. [17]	Review	High levels of screen time and social media use correlates to depression, anxiety, and misperception of body image, addiction, and mental health outcomes.
Depression Sleep Anxiety	Hoge E. et al. [18]	Review	The more time adolescents spend on smartphone, the higher levels of depression, insomnia, and anxiety are found one year later.
Depression	Hoare E. et al. [19]	Clinical study	Adolescents suffering for depression and mental health impairment in adolescence reported a greater use of social media.
Depression	Ha L. et al. [20]	Clinical study	Swedish adolescents who spent more than 2 h on social media had higher odds of feeling depressed.
Depression Diet Cyberbullying Sleep Sex related problems Online grooming	Chassiakos Y.L.R. et al. [21]	Review	Risks of media device use include obesity, sleep, attention, and learning impairment, illicit substance use, high-risk sexual behaviors, depression, cyberbullying, and compromised privacy and confidentiality.
Depression Cyberbullying Sleep Anxiety Sex related problems Behavioral problems Sight	Maurer B.T. et al. [22]	Review	An increase in digital and social media use relates to physical and mental status impairment in children, including depression, anxiety, cyberbullying, sleep disturbance, behavioral problems, sexting, and myopia.
Depression Psychological problems Anxiety	Keles B. et al. [23]	Review	Time spent on social media, repetitive activities, addictive, or problematic use associated with depression, anxiety, and psychological impairment. Nevertheless, it is not possible to establish whether a causative effect exists.
Depression Addiction	Khalil S.A. et al. [24]	Clinical Study	A percentage of 65.6% of Egyptians adolescents are having internet addiction, especially Facebook addiction (92.8%) and gaming (61.3%). Those affected by Facebook addiction are at risk of dysthymia.
Depression Cyberbullying Body image	Richards D. et al. [25]	Review	Social media overuse impacts on mental health, self-esteem, and wellbeing.
Depression Cyberbullying Sleep Sex related problems	Hadjipanayis A. et al. [26]	Review	Social media use facilitates socialization, communication, learning skills, and may positively affect education. Potential risks include cyberbullying, Facebook depression, sleep disturbances, and sexting.
Depression Cyberbullying	Hamm M.P. et al. [27]	Review	Cyberbullying and depression correlate with a regular and constant social media use.
Depression Cyberbullying	Carpenter L.M. et al. [28]	Review	Internet, mobile devices, and social networking sites link to mental health impairment and cyberbullying.
Depression Cyberbullying	Aboujaoude E. et al. [29]	Review	Internet penetrance and connectivity are strictly related to cyberbullying and altered mental status.
Depression	Listernick Z.I. et al. [30]	Review	Depressive symptoms increased during COVID-19 pandemic era. Risk factors include social isolation, family stress, and social media overuse.
Depression Sleep Anxiety	Armitage R.C. et al. [31]	Letter to editor	The degree of social media usage in children correlates with depression, anxiety, and perceived stress level. Bedtime access to and use of mobile devices is significantly associated with inadequate sleep in terms of quality and quantity.
Depression Anxiety	Caffo E. et al. [32]	Review	Many factors including isolation, excessive social media use, and parental stress worsened mental status health during COVID-19 era.
Depression Psychological problems Anxiety	Chen I.H. et al. [33]	Review	During school closure in COVID-19 pandemic smartphone and social media use increased. An increase of 15–30 min daily negatively influenced mental health status in children.

**Table 3 ijerph-19-09960-t003:** Social media and diet.

Domains	Reference	Type of Publication	Highlighted
Depression Diet Cyberbullying Sleep Sex related problems Online grooming	Chassiakos Y.L.R. et al. [21]	Review	Risks of media device use include obesity, sleep, attention, and learning impairment, illicit substance use, high-risk sexual behaviors, depression, cyberbullying, and compromised privacy and confidentiality.
Diet	Théodore F.L. et al. [34]	Clinical Study	Digital Marketing represents a major threat for children and adolescents in Mexico, because of its persuasive techniques.
Diet Dental Caries	Radesky J. et al. [35]	Clinical Study	Exposure to advertising is associated with unhealthy behaviors. Children are uniquely vulnerable to the persuasive effects of advertising because of immature critical thinking skills and impulse inhibition. Ads also promote intake of foods that contribute to dental caries.
Diet	Folkvord F. et al. [36]	Review	Unhealthy food is advertised intensively on several media platforms that are increasingly used by children. This contributes to the obesity epidemic.
Diet	Sacks G. et al. [37]	Clinical Study	Exposure to the marketing of unhealthy products, on social media is associated with a higher risk of related unhealthy behaviors. Analysis of the advertising policies of the 16 largest social media platforms proved them ineffective in protecting children and adolescents from exposure to the digital marketing of unhealthy food.
Diet	Tan L. et al. [38]	Clinical Study	Unhealthy food marketing to children is a key risk factor for childhood obesity. Analysis of ads encountered in YouTube videos targeted at children revealed that food and beverage ads appeared most frequently with more than half of these promoting noncore or unhealthy foods.
Diet	Murphy G. et al. [39]	Clinical Study	Adolescents respond more positively to unhealthy food advertising compared to healthy food or non-food advertising.
Diet	Lutfeali S. et al. [40]	Clinical Study	Heavy social media users (>3 h/day) were 6.366 times more willing to comment on ads compared to light users (*p* < 0.001).
Diet	Khan M. et al. [41]	Clinical Study	The food industry has intensified online advertising focused on children during COVID-19 pandemic, helping the widespread of weight gain.
Diet	Khajeheian D. et al. [42]	Clinical Study	Students, in primary school and high school, who spend more time using social media, exhibit a greater increase in BMI.
Diet Sleep	Mazur A. et al. [43]	Clinical Study	Obesity correlated to junk food advertisement and a more sedentary lifestyle promoted by social media use. Poor or deregulated sleep affects the regulation of energy balance representing a risk factor for childhood obesity.
Diet	Custers K. et al. [44]	Review	Raising in the presence of pro-eating disorder content on websites and social media, which correlates of eating disturbances.
Diet Cyberbullying Psychological problems Body image Physical activity	Borzekowski D.L.G. et al. [45]	Review	Constant media access and exposure to unhealthy and risky media messages may increase the interactions, facilitating cyberbullying and exacerbating body image apprehension promoting poor nutrition, psychological problems, and leading to a more sedentary lifestyle.
Diet Body image	Moorman E.L. et al. [46]	Clinical Study	Greater use the internet sources for nutritional information is related to greater disordered eating.
Diet Sleep Online grooming	Purves R.I. et al. [47]	Clinical Study	Alcohol brands on social media portray drinking identities, appealing for young adolescent and with the potential risk to peer group acceptance.

**Table 4 ijerph-19-09960-t004:** Social media and cyberbullying.

Domains	Reference	Type of Publication	Highlighted
Depression Diet Cyberbullying Sleep Sex related problems Online grooming	Chassiakos Y.L.R. et al. [21]	Review	Risks of media device use include obesity, sleep, attention, and learning impairment, illicit substance use, high-risk sexual behaviors, depression, cyberbullying, and compromised privacy and confidentiality.
Depression Cyberbullying Sleep Anxiety Sex related problems Behavioral problems Sight	Maurer B.T. et al. [22].	Review	An increase in digital and social media use relates to physical and mental status impairment in children, including depression, anxiety, cyberbullying, sleep disturbance, behavioral problems, sexting, and myopia.
Depression Cyberbullying Body image	Richards D. et al. [25]	Review	Social media overuse impacts on mental health, self-esteem, and wellbeing.
Depression Cyberbullying Sleep Sex related problems	Hadjipanayis A. et al. [26]	Review	Social media use facilitates socialization, communication, learning skills, and may positively affect education. Potential risks include cyberbullying, Facebook depression, sleep disturbances, and sexting.
Depression Cyberbullying	Hamm M.P. et al. [27]	Review	Cyberbullying and depression correlate with a regular and constant social media use.
Depression Cyberbullying	Carpenter L.M. et al. [28]	Review	Internet, mobile devices, and social networking sites link to mental health impairment and cyberbullying.
Depression Cyberbullying	Aboujaoude E. et al. [29]	Review	Internet penetrance and connectivity are strictly related to cyberbullying and altered mental status.
Diet Cyberbullying Psychological problems Body image Physical activity	Borzekowski D.L.G. et al. [45]	Review	Constant media access and exposure to unhealthy and risky media messages may increase the interactions, facilitating cyberbullying and exacerbating body image apprehension promoting poor nutrition, psychological problems, and leading to a more sedentary lifestyle.
Cyberbullying	Wise J. et al. [48]	Letter to Editor	Potential negative effects of social media include damage to sleep patterns, cyberbullying, and online grooming.
Cyberbullying Psychological problems Sleep Addiction Behavioral problems Physical activity Sight	Bozzola E. et al. [49]	Review	Cyberbullying, sleep impairment, psychological problems, addiction, musculoskeletal disorders, and eye problems are among the risks of media device use in adolescence.
Cyberbullying Sex related problems Body image	Shah J. et al. [50]	Review	Increased social media usage correlates with decreased self-esteem and body satisfaction, increment of cyberbullying, and exposure to pornographic material and risky sexual behaviors.
Cyberbullying Addiction Sex related problems	O’Keeffe G.S. et al. [51]	Editorial	Risks of social media overuse include sexting, cyberbullying, privacy issues, and Internet addiction, all of which may present with vague health symptoms.
Cyberbullying Psychological problems Addiction	Nagata J.M. et al. [52]	Editorial	More than 7% of adolescents have problematic media use and addiction to social media. Problematic media use is associated with cyberbullying, poor outcomes in life satisfaction, and mental health.
Cyberbullying	Marengo N. et al. [53]	Clinical study	The risk of cyber-victimization is higher in case of problematic social media use and in female gender.
Cyberbullying	Uludasdemir D. et al. [54]	Clinical study	Having daily access to the Internet and the sharing of gender on social media increased the likelihood of cyber victimization.

**Table 5 ijerph-19-09960-t005:** Social media and psychological problems.

Domains	Reference	Type of Article	Highlighted
Depression Psychological problems Addiction Anxiety Body image	Mougharbel F. et al. [17]	Review	High levels of screen time and social media use correlates to depression, anxiety, misperception of body image, and mental health outcomes.
Depression Psychological problems Anxiety	Keles B. et al. [23]	Review	Time spent on social media, repetitive activities, addictive, or problematic use associated with depression, anxiety, and psychological impairment. Nevertheless, it is not possible to establish whether a causative effect exists.
Depression Psychological problems Anxiety	Chen I.H. et al. [33]	Review	During school closure in COVID-19 pandemic smartphone and social media use increased. An increase of 15–30 min daily negatively affected mental health status in children.
Diet Cyberbullying Psychological problems Body image Physical activity	Borzekowski D.L.G. et al. [45]	Review	Constant media access and exposure to unhealthy and risky media messages may increase the interactions, facilitating cyberbullying and exacerbating body image apprehension promoting poor nutrition, psychological problems, and leading to a more sedentary lifestyle.
Cyberbullying Psychological problems Sleep Addiction Behavioral problems Physical activity Sight	Bozzola E. et al. [49]	Review	Cyberbullying, sleep impairment, psychological problems, addiction, musculoskeletal disorders, and eye problems are among the risks of media device use in adolescence.
Cyberbullying Psychological problems Addiction	Nagata J.M. et al. [52]	Editorial	More than 7% of adolescents have problematic media use and addiction to social media. Problematic media use is associated with cyberbullying, poor outcomes in life satisfaction, and mental health.
Psychological problems	Favotto L. et al. [55]	Clinical study	Children with low family communication have high levels of media use and loneliness.
Psychological problems	Boer M. et al. [56]	Clinical study	Data among 154,981 adolescents of the world, described that problematic media use is associated with lower well-being.
Psychological problems Sleep Physical activity	Buda G. et al. [57]	Clinical study	Problematic social media use correlates with about two times higher odds for worse sleep quality and lower life satisfaction, and it is related to lower levels of vigorous physical activity in girls.
Psychological problems	Mc Dool E. et al. [58]	Clinical study	Among 6300 English students, internet use is negatively associated with feel about appearance, especially in girls.
Psychological problems	Twigg L. et al. [59]	Clinical study	Higher levels of social media use are associated with lower happiness, especially in girls.
Psychological problems	Walsh S.D. et al. [60]	Clinical study	Problematic social media use such as substance use, bullying, and low social support, have been identified as clusters of risk for children mental health.
Psychological problems Sleep	Sümen A. et al. [61]	Clinical study	Social media addiction in school students is related with lower communication among families, loneliness, emotional problems, attention deficit, peer problems, and it decreases students’ sleep efficiency.
Psychological problems Headache	Marino C. et al. [62]	Clinical study	Adolescent problematic Internet users have higher levels of somatic symptoms such as headaches and psychological consequences of social media use such as loss of control and relational problems with family and friends.

**Table 6 ijerph-19-09960-t006:** Social media and sleep.

Domains	Reference	Type of Article	Highlighted
Depression Sleep Anxiety	Hoge E. et al. [18]	Review	The more time adolescents spend on smartphone, the higher levels of depression, insomnia, and anxiety are found one year later.
Depression Diet Cyberbullying Sleep Sex related problems Online grooming	Chassiakos Y.L.R. et al. [21]	Review	Risks of media device use include obesity, sleep, attention, and learning impairment, illicit substance use, high-risk sexual behaviors, depression, cyberbullying, and compromised privacy and confidentiality.
Depression Cyberbullying Sleep Anxiety Sex related problems Behavioral problems Sight	Maurer B.T. et al. [22].	Review	An increase in digital and social media use relates to physical and mental status impairment in children, including depression, anxiety, cyberbullying, sleep disturbance, behavioral problems, sexting, and myopia.
Depression Cyberbullying Sleep Sex related problems	Hadjipanayis A. et al. [26]	Review	Social media use facilitates socialization, communication, learning skills, and may positively influence education. Potential risks include cyberbullying, Facebook depression, sleep disturbances, and sexting.
Depression Sleep Anxiety	Armitage R.C. et al. [31]	Letter	The degree of social media usage in children correlates with depression, anxiety, and perceived stress level. Bedtime access to and use of mobile devices is significantly associated with inadequate sleep in terms of quality and quantity.
Diet Sleep	Mazur A. et al. [43]	Clinical study	Obesity correlated to junk food advertisement and a more sedentary lifestyle promoted by social media use. Poor or deregulated sleep affects the regulation of energy balance representing a risk factor for childhood obesity.
Diet Sleep Online grooming	Purves R.I. et al. [47]	Letter	Potential negative effects of social media include damage to sleep patterns, cyberbullying, and online grooming.
Cyberbullying Psychological problems Sleep Addiction Behavioral problems Physical activity Sight	Bozzola E. et al. [49]	Review	Cyberbullying, sleep impairment, psychological problems, addiction, musculoskeletal disorders, and eye problems are among the risks of media device use in adolescence.
Psychological problems Sleep Physical activity	Buda G. et al. [57]	Clinical study	Problematic social media use correlates with about two times higher odds for worse sleep quality and lower life satisfaction, and it is related to lower levels of vigorous physical activity in girls.
Psychological problems Sleep	Sümen A. et al. [61]	Clinical study	Social media addiction in school students is related with lower communication among families, loneliness, emotional problems, attention deficit, peer problems, and it decreases students’ sleep efficiency.
Sleep	Varghese N.E. et al. [63]	Clinical study	Exposure to media device and social media is significantly associated with adolescent sleep-onset difficulties.
Sleep Behavioral problems	Guerrero M.D. et al. [64]	Clinical study	Time spent on screen has been associated to sleeping problems, especially sleep duration, and with problematic behaviors, higher internalizing, and externalizing symptoms.
Sleep	Lund L. et al. [65]	Review	Relationship between social media use, late sleep onset, sleep quality, and duration.

**Table 7 ijerph-19-09960-t007:** Social media and addiction.

Domains	Reference	Type of Article	Highlighted
Depression Addiction Anxiety	Chiu M. et al. [15]	Clinical Study	Social media increased use correlates to Emergency Department visits for mental illness, including depression, addiction, and anxiety.
Depression Psychological problems Addiction Anxiety Body image	Mougharbel F. et al. [17]	Review	High levels of screen time and social media use correlates to depression, anxiety, and misperception of body image, addiction, and mental health outcomes.
Depression Addiction	Khalil S.A. et al. [24].	Clinical Study	A percentage of 65.6% of Egyptians adolescents are having internet addiction, especially Facebook addiction (92.8%) and gaming (61.3%). Those affected by Facebook addiction are at risk of dysthymia.
Cyberbullying Psychological problems Sleep Addiction Behavioral problems Physical activity Sight	Bozzola E. et al. [49]	Review	Cyberbullying, sleep impairment, psychological problems, addiction, musculoskeletal disorders, and eye problems are among the risks of media device use in adolescence.
Cyberbullying Addiction Sex related problems	O’Keeffe G.S. et al. [51]	Editorial	Risks of social media overuse include sexting, cyberbullying, privacy issues, and Internet addiction, all of which may present with vague health symptoms.
Cyberbullying Psychological problems Addiction	Nagata J.M. et al. [52]	Editorial	More than 7% of adolescents have problematic media use and addiction to social media. Problematic media use is associated with cyberbullying, poor outcomes in life satisfaction, and mental health.
Addiction	Hawi N.S. et al. [66]	Clinical Study	The Digital Addiction Scale for Children was validated on 822 participants, to assess the behavior of children 9 to 12 years old in association with video gaming, social media, and texting. Females are more susceptible to social media addiction.
Addiction	Turhan P. et al. [67]	Clinical Study	Among a group of 93 adolescents with substance abuse, social media addiction and gaming disorders have been documented more than control group.
Addiction	Emond A.M. et al. [68]	Review	Gambling and gaming addiction are emerging problems in children and adolescents. Children are exposed to gambling adverts using media device and television. Moreover, social media sometimes promotes gambling.
Addiction	Unger J.B. et al. [69]	Clinical Study	Tobacco use in adolescents correlates to tobacco content on social media. In particular, adolescents with more tobacco tweets were more likely to use cigarettes. Advertising messages about tobacco shared trough social media have been connected to tobacco use in adolescents.

**Table 8 ijerph-19-09960-t008:** Social media and anxiety.

Domains	Reference	Type of Publication	Highlighted
Depression Addiction Anxiety	Chiu M. et al. [15]	Clinical Study	Social media increased use correlates to Emergency Department visits for mental illness, including depression, addiction, and anxiety.
Depression Anxiety	Rutter L.A. et al. [16]	Clinical Study	Social media use correlates with depressive symptoms, anxiety, and loneliness. Physical activity negatively correlates with depression.
Depression Psychological problems Addiction Anxiety Body image	Mougharbel F. et al. [17]	Review	High levels of screen time and social media use correlates to depression, anxiety, and misperception of body image, addiction, and mental health outcomes.
Depression Sleep Anxiety	Hoge E. et al. [18]	Review	The more time adolescents spend on smartphone, the higher levels of depression, insomnia, and anxiety are found one year later.
Depression Cyberbullying Sleep Anxiety Sex related problems Behavioral problems Sight	Maurer B.T. et al. [22]	Review	An increase in digital and social media use relates to physical and mental status impairment in children, including depression, anxiety, cyberbullying, sleep disturbance, behavioral problems, sexting, and myopia.
Depression Psychological problems Anxiety	Keles B. et al. [23]	Review	Time spent on social media, repetitive activities, addictive or problematic use associated with depression, anxiety, and psychological impairment. Nevertheless, it is not possible to establish whether a causative effect exists.
Depression Sleep Anxiety	Armitage R.C. et al. [31]	Letter	The degree of social media usage in children correlates with depression, anxiety, and perceived stress level. Bedtime access to and use of mobile devices is significantly associated with inadequate sleep in terms of quality and quantity.
Depression Anxiety	Caffo E. et al. [32]	Review	Many factors including isolation, excessive social media use and parental stress worsened mental status health during COVID-19 era.
Depression Psychological problems Anxiety	Chen I.H. et al. [33]	Review	During school closure in COVID-19 pandemic smartphone and social media use increased. An increase of 15–30 min daily negatively influenced mental health status in children.
Anxiety	Muzaffar N. et al. [70]	Clinical Study	Increased anxiety correlates with increased Facebook use and repetitive behavior on social media among adolescents.

**Table 9 ijerph-19-09960-t009:** Social media and sex related problems.

Domains	Reference	Type of Publication	Highlighted
Depression Diet Cyberbullying Sleep Sex related problems Online grooming	Chassiakos Y.L.R. et al. [21]	Review	Risks of media device use include obesity, sleep, attention, and learning impairment, illicit substance use, high-risk sexual behaviors, depression, cyberbullying, and compromised privacy and confidentiality.
Depression Cyberbullying Sleep Anxiety Sex related problems Behavioral problems Sight	Maurer B.T. et al. [22].	Review	An increase in digital and social media use relates to physical and mental status impairment in children, including depression, anxiety, cyberbullying, sleep disturbance, behavioral problems, sexting, and myopia.
Depression Cyberbullying Sleep Sex related problems	Hadjipanayis A. et al. [26]	Review	Social media use facilitates socialization, communication, learning skills, and may positively affect education. Potential risks include cyberbullying, Facebook depression, sleep disturbances, and sexting.
Cyberbullying Sex related problems Body image	Shah J. et al. [50]	Review	Increased social media usage correlates with decreased self-esteem and body satisfaction, increment of cyberbullying, exposure to pornographic material, and risky sexual behaviors.
Cyberbullying Addiction Sex related problems	O’Keeffe G.S. et al. [51]	Editorial	Risks of social media overuse include sexting, cyberbullying, privacy issues, and Internet addiction, all of which may present with vague health symptoms.
Sex related problems	Gazendam N. et al. [71]	Clinical Study	Sexual activity has been described in a sample of 7882 Canadian students. An increase of sexual activity has been observed in both girls and boys using media. A great social media use has been connected to the strongest association with early sexual activity for girls (RR = 1.42, 95% CI: 1.01–1.47).
Sex related problems	Wana G. et al. [72]	Clinical Study	Social media use has been described as a risk factor for sexual behavior. About 7% of adolescents use media to watch pornographic content.
Sex related problems Body image Physical activity Sight Headache	Solecki S. et al. [73]	Clinical Study	Addictive, smart phone activities of youth is directly affecting their perception of the body, also causing physical problems, eye diseases, headache, and exposure to unwanted sexual material online.
Sex related problems	Collins R.L. et al. [74]	Clinical Study	Traditional media and social media use among adolescents are related to sexual activities and behavior. Video games contain sexual contents. New technologies facilitate pornography access among young.

**Table 10 ijerph-19-09960-t010:** Social media and behavioral problems.

Domains	Reference	Type of Publication	Highlighted
Depression Cyberbullying Sleep Anxiety Sex related problems Behavioral problems	Maurer B.T. et al. [22].	Review	An increase in digital and social media use relates to physical and mental status impairment in children, including depression, anxiety, cyberbullying, sleep disturbance, behavioral problems, sexting, and myopia.
Cyberbullying Psychological problems Sleep Addiction Behavioral problems Physical activity Sight	Bozzola E. et al. [49]	Review	Cyberbullying, sleep impairment, psychological problems, addiction, musculoskeletal disorders, and eye problems are among the risks of media device use in adolescence.
Sleep Behavioral problems	Guerrero M.D. et al. [64]	Clinical Study	Time spent on screen has been associated to sleeping problems, especially sleep duration, and with problematic behaviors, higher internalizing, and externalizing symptoms.
Behavioral problems	McNamee P. et al. [75].	Clinical Study	Excessive time of media use has a strong association with emotional distress and worse behavioral outcomes.
Behavioral problems	Okada S. et al. [76].	Clinical Study	Association between hours of media use and behavioral problems has been documented among children aged 9–10 years old, in Japan.
Behavioral problems	Tahir A. et al. [77].	Clinical Study	Regression analysis predicted a strong positive association of exposure to violent social/electronic media content with level of aggression of adolescents (β = 0.43).
Behavioral problems	Deslandes S.F. et al. [78].	Clinical Study	Online challenges are a powerful communicative resource but can involve potential self-inflicted injuries to participants, with risks ranging from minor to lethal.

**Table 11 ijerph-19-09960-t011:** Social media and body image.

Domains	Reference	Type of Publication	Highlighted
Depression Psychological problems Addiction Anxiety Body image	Mougharbel F. et al. [17]	Review	High levels of screen time and social media use correlates to depression, anxiety, and misperception of body image, addiction, and mental health outcomes.
Depression Cyberbullying Body image	Richards D. et al. [25]	Review	Social media overuse impacts on mental health, self-esteem, and wellbeing.
Diet Cyberbullying Psychological problems Body image Physical activity	Borzekowski D.L.G. et al. [45]	Review	Constant media access and exposure to unhealthy and risky media messages may increase the interactions, facilitating cyberbullying and exacerbating body image apprehension promoting poor nutrition, psychological problems, and leading to a more sedentary lifestyle.
Diet Body image	Moorman E.L. et al. [46]	Clinical Study	Greater use the internet sources for nutritional information is related to greater disordered eating.
Cyberbullying Sex related problems Body image	Shah J. et al. [50]	Review	Increased social media usage correlates with decreased self-esteem and body satisfaction, increment of cyberbullying, exposure to pornographic material, and risky sexual behaviors.
Sex related problems Body image Physical activity Sight Headache	Solecki S. et al. [73]	Review	Addictive, smart phone activities of youth is directly affecting their perception of the body, also causing physical problems, eye diseases, headache, and exposure to unwanted sexual material online.

**Table 12 ijerph-19-09960-t012:** Social media and physical activity.

Domains	Reference	Type of Publication	Highlighted
Diet Cyberbullying Psychological problems Body image Physical activity	Borzekowski D.L.G. et al. [45]	Review	Constant media access and exposure to unhealthy and risky media messages may increase the interactions, facilitating cyberbullying, and exacerbating body image apprehension promoting poor nutrition, psychological problems, and leading to a more sedentary lifestyle.
Cyberbullying Psychological problems Sleep Addiction Behavioral problems Physical activity Sight	Bozzola E. et al. [49]	Review	Cyberbullying, sleep impairment, psychological problems, addiction, musculoskeletal disorders, and eye problems are among the risks of media device use in adolescence.
Psychological problems Sleep Physical activity	Buda G. et al. [57]	Clinical Study	Problematic social media use correlates with about two times higher odds for worse sleep quality and lower life satisfaction, and it is related to lower levels of vigorous physical activity in girls.
Sex related problems Body image Physical activity Sight Headache	Solecki S. et al. [73]	Review	Addictive, smart phone activities of youth is directly influencing their perception of the body, also causing physical problems, eye diseases, headache, and exposure to unwanted sexual material online.
Physical activity	Kemp B.J. et al. [79]	Clinical Study	Australian children between 11 y and 13 y who had a larger increase in social media use had lower participation in overall physical activity.

**Table 13 ijerph-19-09960-t013:** Social media and online grooming.

Domains	Reference	Type of Publication	Highlighted
Depression Diet Cyberbullying Sleep Sex related problems Online grooming	Chassiakos Y.L.R. et al. [21]	Review	Risks of media device use include obesity, sleep, attention, and learning impairment, illicit substance use, high-risk sexual behaviors, depression, cyberbullying, and compromised privacy and confidentiality.
Diet Sleep Online grooming	Purves R.I. et al. [47]	Letter	Potential negative effects of social media include damage to sleep patterns, cyberbullying, and online grooming.
Online grooming	Forni G. et al. [80]	Review	This review describes the online grooming phenomenon, victim and perpetrators characteristics, and the importance to implement attention on this problem with preventive measures.

**Table 14 ijerph-19-09960-t014:** Social media and sight.

Domains	Reference	Type of Publication	Highlighted
Depression Cyberbullying Sleep Anxiety Sex related problems Behavioral problems Sight	Maurer B.T. et al. [22].	Review	An increase in digital and social media use relates to physical and mental status impairment in children, including depression, anxiety, cyberbullying, sleep disturbance, behavioral problems, sexting, and myopia.
Cyberbullying Psychological problems Sleep Addiction Behavioral problems Physical activity Sight	Bozzola E. et al. [49]	Review	Cyberbullying, sleep impairment, psychological problems, addiction, musculoskeletal disorders, and eye problems are among the risks of media device use in adolescence.
Sex related problems Body image Physical activity Sight Headache	Solecki S. et al. [73]	Review	Addictive, smart phone activities of youth is directly influencing their perception of the body, also causing physical problems, eye diseases, headache, and exposure to unwanted sexual material online.

**Table 15 ijerph-19-09960-t015:** Social Media and headache.

Domains	Reference	Type of Publication	Highlighted
Psychological problems Headache	Marino C. et al. [62]	Clinical Study	Adolescent problematic Internet users have higher levels of somatic symptoms such as headaches and psychological consequences of social media use such as loss of control and relational problems with family and friends.
Sex related problems Body image Physical activity Sight Headache	Solecki S. et al. [73]	Review	Addictive, smart phone activities of youth is directly influencing their perception of the body, also causing physical problems, eye diseases, headache, and exposure to unwanted sexual material online.
Headache	Çaksen H. et al. [81]	Review	Abuse of electronic screens more than 2 h contributes to the chance of reporting headache.

**Table 16 ijerph-19-09960-t016:** Social media and dental caries.

Domains	Reference	Type of Publication	Highlighted
Diet Dental Caries	Radesky J. et al. [35]	Clinical Study	Exposure to advertising is associated with unhealthy behaviors. Children are uniquely vulnerable to the persuasive effects of advertising because of immature critical thinking skills and impulse inhibition. Ads also promote intake of foods that contribute to dental caries.
Dental Caries	Almoddahi D. [82]	Clinical Study	Study conducted in England, Wales, and Northern Ireland. Excessive internet use is associated to dental caries, and this could be mediated by health behaviors.

## Data Availability

Data available at Dr Bozzola’s study.

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
