# Peer review of "The Use of Social Media in Children and Adolescents: Scoping Review on the Potential Risks"

_ijerph, 2022, doi:10.3390/ijerph19169960_

Round 1

Reviewer 1 Report

All the amendments have been made.

Reviewer 2 Report

The autohors have responde correctly to my questions.

The paper has gainde a lot of quality with the suggestions of my colleagues.

It looks lie another article.

Authors should delete the initials of the authors' names in the tables.

This manuscript is a resubmission of an earlier submission. The following is a list of the peer review reports and author responses from that submission.

Round 1

Reviewer 1 Report

Given that the literature review section has been incorporated into the 'Introduction', I think that there is a need for an extra paragraph to connect the introduction to the 'materials and methods section'. In parallel, the article refers to potential risks. Therefore, I think that the risk definition should be included in the methodology. Taking into consideration that the risk definition is heavily dependent on the context and the area of application, it is of utmost importance to define the risk context in order to be able to view the respective factors as potential risk factors. Additionally, the results' outcome is not fully in connection with risks. In other words, authors should focus on clarifying why the factors that have been referred to could be considered as potential risk factors. 

Reviewer 2 Report

There are many problems with this manuscript that need to be improved.

IN GENERAL

The overall writing skills of the manuscript need to be improved.

ABSTRACT

Lack of attraction for readers to read further.

INTRODUCTION

Lack of a comprehensive review of the impact of social media on children and adolescents.

MATERIALS AND METHODS

Lack of detailed step-by-step and process descriptions, including: literature acquisition steps (strategies), exclusion/inclusion criteria, data extraction, potential bias risk and statistical methods

RESULTS

In all the systematic review articles included in the study, no commonalities or differences in findings were collated.

CONCLUSION

Lack of comparative discussion focusing on the statistical results of the systematic review articles.

Reviewer 3 Report

Introduction

In the introduction they focus on the effects of the pandemic, but before and after the pandemic there is already evidence of the problems of excessive use of the internet in general and of social networks, video games, pornography, .... in children and adolescents.

The introduction is very weak. More work needs to be done on it. There are no references to international studies.

At what age do you mean pediatric population?

They should define what they mean by social media meant as forms of electronic communication

…of the 922 works identified, all abstracts were analyzed, and 832 records were excluded. Around 66% of the excluded records were dealing with other topics (e.g., vaccines, promoting health by social media, social networks meant as real social interactions, social lockdown during SARS-CoV2 period), a percentage of 28% of records corresponded to a wrong population: mostly parents, pregnant women, young adults, or children with pathologies (e.g., ADHD). About 6% of the excluded studies used social media tools to recruit people in their studies or to deliver questionnaires…

It is obvious, from my point of view, that the search keywords are not the right ones. Going from 1005 to 68 articles can only indicate that the inclusion criteria have not been well used.

Discussion.

The corresponding tables with the most relevant data for each article are missing in the results. And the tables presented are not those indicated for a PRISMA review.

There is no reference to COVID or to the effects of the pandemic in the title or in the keywords, but the depression table (among others) shows before and after COVID.

Conclusion

While the conclusions are interesting and have a very important social relevance. The authors should provide proposals to reduce the negative effects of the excessive use of social media.

The authors should add a section on the limitations of this study.